# Friction and Wear Monitoring Methods for Journal Bearings of Geared Turbofans Based on Acoustic Emission Signals and Machine Learning

**Noushin Mokhtari** [1,*], **Jonathan Gerald Pelham** [2], **Sebastian Nowoisky** [3],
**José-Luis Bote-Garcia** [1] **and Clemens Gühmann** [1]

[1] Chair of Electronic Measurement and Diagnostic, Department of Energy and Automation Technology, School of Electrical Engineering and Computer Science, Technische Universität Berlin, Straße des 17. Juni 135, 10623 Berlin, Germany; jose-luis.bote-garcia@tu-berlin.de (J.-L.B.-G.); clemens.guehmann@tu-berlin.de (C.G.)

[2] Safety and Accident Investigation Centre, School of Aerospace, Transport and Manufacturing, Cranfield University, College Road, Cranfield, Bedfordshire MK43 0AL, UK; j.g.pelham@cranfield.ac.uk

[3] Rolls-Royce Deutschland Ltd. & Co. KG, Eschenweg 11, 15827 Blankenfelde-Mahlow, Germany; sebastian.nowoisky@rolls-royce.com

* Correspondence: noushin.mokhtari@tu-berlin.de

**Abstract:** In this work, effective methods for monitoring friction and wear of journal bearings integrated in future UltraFan® jet engines containing a gearbox are presented. These methods are based on machine learning algorithms applied to Acoustic Emission (AE) signals. The three friction states: dry (boundary), mixed, and fluid friction of journal bearings are classified by pre-processing the AE signals with windowing and high-pass filtering, extracting separation effective features from time, frequency, and time-frequency domain using continuous wavelet transform (CWT) and a Support Vector Machine (SVM) as the classifier. Furthermore, it is shown that journal bearing friction classification is not only possible under variable rotational speed and load, but also under different oil viscosities generated by varying oil inlet temperatures. A method used to identify the location of occurring mixed friction events over the journal bearing circumference is shown in this paper. The time-based AE signal is fused with the phase shift information of an incremental encoder to achieve an AE signal based on the angle domain. The possibility of monitoring the run-in wear of journal bearings is investigated by using the extracted separation effective AE features. Validation was done by tactile roughness measurements of the surface. There is an obvious AE feature change visible with increasing run-in wear. Furthermore, these investigations show also the opportunity to determine the friction intensity. Long-term wear investigations were done by carrying out long-term wear tests under constant rotational speeds, loads, and oil inlet temperatures. Roughness and roundness measurements were done in order to calculate the wear volume for validation. The integrated AE Root Mean Square (RMS) shows a good correlation with the journal bearing wear volume.

**Keywords:** journal bearing; acoustic emission; machine learning; friction classification; friction localization; run-in wear; long-term wear

## 1. Introduction

An effective means of improving turbofan engine efficiency is to increase the bypass ratio (BPR). The BPR is driven by the fan diameter associated with aerodynamic influences combined with the thermodynamic requirements of the turbine design. Ideally, the fan operates at slow speeds and

the turbine at high rotational speeds. These contradicting requirements can be sorted out by using a planetary gearbox between the components. However, using a gearbox introduces additional failure modes such as journal bearing wear caused by mixed or dry friction. A breakdown of this component could have a negative impact on the product reliability which causes high maintenance costs and downtime. This paper outlines journal bearing monitoring opportunities to address technical diagnosis of the world's most powerful aircraft gearbox.

The Power Gearbox (PGB) is a demonstrator programme in preparation for future UltraFan® design standard jet engine products. In this planetary gearbox monitoring of journal bearings is required. It is a general requirement for an aero engine to enable early fault detection. To meet this requirement, an analysis of the novel power plant architecture was performed using various System Design tools such as Function Failure Mode & Effect Analysis (FFMEA) to identify journal bearing monitoring options. Previous works have already shown the possibility of identifying journal bearing mixed and dry friction by using the acoustic emission (AE) technology [1–4]. Other options such as temperature, friction torque, debris, or position monitoring can not be applied for several reasons e.g., sensor sensitivity or limited design space.

In addition to knowledge of the current state of friction, knowledge of the current state of wear, which means the current loss of material, is also necessary. Some work already focused on the detection of wear using AE analysis [5–9]. However, the literature only contains investigations on pin-disc tribometers, the results of which are supposed to prove the possible applicability to real systems. In most cases, the correlation between wear measurements and AE features is investigated. These investigations are idealized and in most cases the applicability to real applications is not proven. Wear investigations based on AE using grease-lubricated journal bearings were done by Hase [10]. He shows the possibility to detect different modes of wear with AE. The applicability to oil-lubricated journal bearings is not shown.

As already noted a reduction gearbox is integrated between fan and turbine. This gearbox can be designed as star arrangement with a fixed carrier and direct access to measure the bearings. To increase the gear ratio a planetary design is used, with a fixed ring gear and a rotating carrier-shaft. To measure the acoustic emission signal of the journal bearing the sensor must be located on the carrier-shaft and the signal must be transferred to a static location while the shaft is rotating with fan speed $w_c$ [11]. Figure 1 shows a schematic of the PGB design with applied wireless data transfer unit (WDTU).

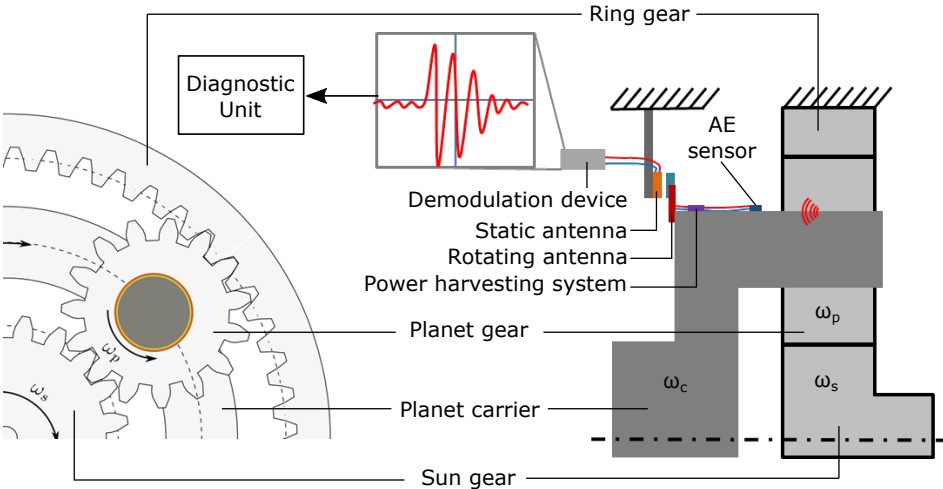

**Figure 1.** Schematic of planetary gearbox and applied Wireless Data Transfer Unit (WDTU).

The environmental conditions in the PGB are very harsh. The WDTU must deal with high vibrations and temperatures above 120 °C, which limits the use of active electronics to measure the AE signal on the rotating part. A solution was identified in [12–14] to transfer the AE signal from the rotating planet to the stationary gearbox for an aviation gearbox application. To demonstrate the

option of detecting the journal bearing mixed friction in a PGB environment it is planned to apply the WDTU on a subscale gearbox [15,16], before a full scale test will be carried out.

With the announced digital strategy of Rolls-Royce to create intelligent engines, further signal processing enables gathering additional information about the UltraFan® power plant [17]. With this option, condition-based maintenance can be done in addition to time-based maintenance. The advantages of this maintenance method are, for example, the extension of operating time and the advance planning of maintenance. A reliable diagnosis and prognosis system is essential for the application of such a condition-based maintenance. For this purpose, machine learning algorithms can be applied to the acquired sensor signals. Figure 2 shows a pattern recognition scheme for diagnosis of different friction states with additional possibility to predict the actual wear condition via regression analysis.

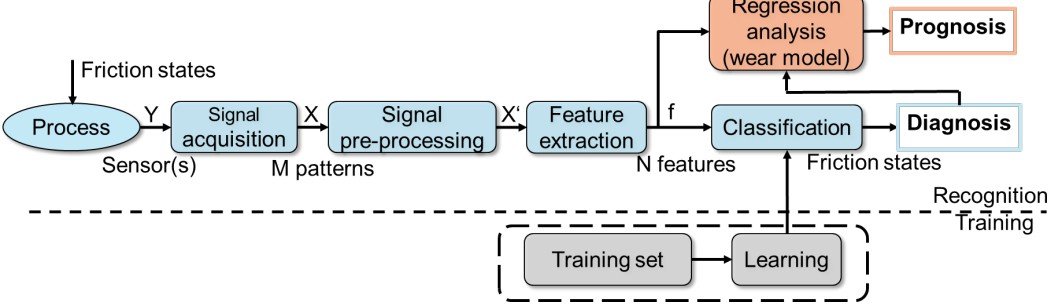

**Figure 2.** Pattern recognition chain for diagnosis of friction states and prognosis of wear condition.

The following research activities can be divided in journal bearing friction investigations such as friction states classification or friction localization, and journal bearing wear investigations including run-in and long-term wear based on a subscale test rig located at Technische Universität Berlin. For the first purpose machine learning methods were applied to classify the three basic journal bearing friction states fluid, mixed, and dry friction. As a result of these first investigations the location on the bearing surface at which mixed friction is occurring can be determined [18]. Furthermore, it will be investigated whether the extracted AE features are applicable for run-in and long-term wear monitoring of oil-lubricated hydrodynamic journal bearings. Run-in wear is generated during removal of the roughness peaks and long-term wear changes the round shape of the journal bearing. Above a certain degree of shape deviation, the supporting lubrication film can no longer be formed, so that the end of lifetime is reached [19]. The roughness and roundness of the journal bearing was measured for validation.

## 2. Experimental Methods

In this section, experimental methods including the used materials, test rigs and experimental procedures are presented.

### 2.1. Materials

The test item used in this work was a journal bearing bush made of the cast material red brass (Cu Sn7 Zn4 Pb7-C). This material made of a copper-zinc-tin alloy offers good sliding and dry running properties and is relatively resistant to wear and cavitation. This material was chosen because it is used in many applications such as turbines, piston engines and gearboxes. The bearing width was 40 mm for the results shown in Sections 3.1–3.3. For the results shown in Section 3.4 the bearing width was reduced to 25 mm in order to increase the specific pressure $p$.

A hardened shaft made of 16MnCr5 steel was used as sliding partner. Manganese chromium alloys are used when high wear resistance is required. As friction and wear are investigated in this

paper, this material was used to avoid failure of the sliding partner. Figure 3 shows the test item and the sliding partner with their significant characteristics.

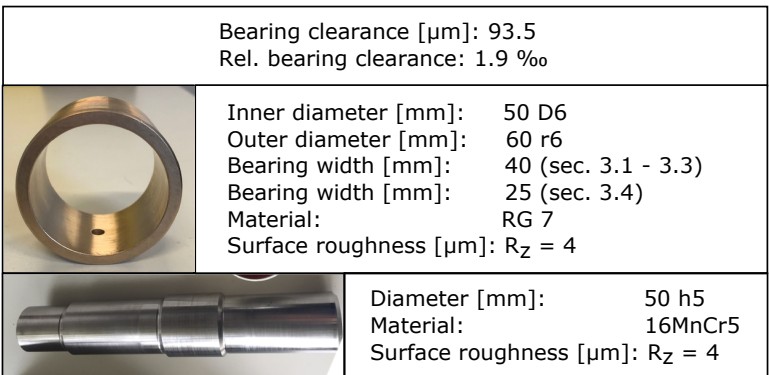

| Bearing clearance [µm]: 93.5 |
| Rel. bearing clearance: 1.9 ‰ |

| Inner diameter [mm]: | 50 D6 |
| Outer diameter [mm]: | 60 r6 |
| Bearing width [mm]: | 40 (sec. 3.1 - 3.3) |
| Bearing width [mm]: | 25 (sec. 3.4) |
| Material: | RG 7 |
| Surface roughness [µm]: $R_Z = 4$ |

| Diameter [mm]: | 50 h5 |
| Material: | 16MnCr5 |
| Surface roughness [µm]: $R_Z = 4$ |

**Figure 3.** Test item (**top**) and sliding partner (**bottom**) used for friction and wear experiments.

## 2.2. Journal Bearing Test Rigs

Two test rigs were used for the friction and wear investigations: the small journal bearing test rig (STR) described in Section 2.2.1, and the temperature-controlled journal bearing test rig (TCTR) described in Section 2.2.2. Table 1 shows the linking of the different test rigs to the respective section of results.

**Table 1.** Linking test rigs to the respective section of results.

| Test Rig | Results in Section |
|----------|--------------------|
| STR | Section 3.1.1–3.1.4/Section 3.2/Section 3.3 |
| TCTR | Section 3.1.5/Section 3.4 |

### 2.2.1. Small Journal Bearing Test Rig (STR)

The STR was specifically developed for this application. It was designed to prove concepts rather than to be representative of the operating environment. Figure 4 shows this test rig and the main operating parameters.

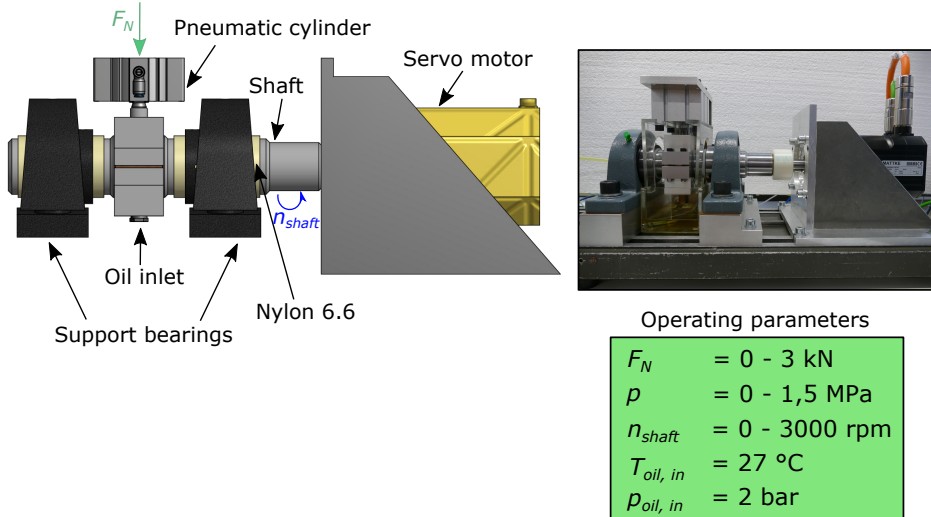

Operating parameters

| $F_N$ | = 0 - 3 kN |
| $p$ | = 0 - 1,5 MPa |
| $n_{shaft}$ | = 0 - 3000 rpm |
| $T_{oil, in}$ | = 27 °C |
| $p_{oil, in}$ | = 2 bar |

**Figure 4.** Small journal bearing test rig (STR) and operating parameters.

The principle is that a radial load $F_N$ is applied to the stationary journal bearing, pressing it against the rotating shaft. If the load is higher than the hydrodynamic lubricating film pressure, which

is generated between the journal bearing and the shaft, the two sliding partners come into contact and mixed or dry friction occurs, otherwise fluid friction is generated.

The shaft is driven by a speed-controlled servo motor from Mattke (HSR0530/ L4-60-P). Control is achieved via a servo controller (Mattke MDR 2300 SE). A pneumatic cylinder from Festo (ADVC-63-10-A-P) is used to generate the radial load $F_N$. Two support bearings are fitted to the right and left of the journal bearing to allow easy mounting and dismounting. Two nylon 6.6 rings have been fitted between the shaft and the support bearings to dampen interference signals generated from the support bearings. The bearing back consists of two parts, which are fixed with screws, to allow easy replacement of the journal bearing. The sliding surfaces were lubricated with a mineral oil of ISO VG class 10. The oil had an approximately constant temperature of $T_{oil,in} = 27\,°C$ during the tests. As heating of the oil was not possible at this time, this low viscosity became necessary in order to be able to generate all three friction conditions with the given speed and load limits. The oil supply pressure $p_{oil,in}$ was set to 2 bar.

### 2.2.2. Temperature-Controlled Journal Bearing Test Rig (TCTR)

To be able to set defined values of oil temperature the STR was equipped with a temperature control system. This became necessary because journal bearing friction states are not only influenced by rotational speed and load variations, but also by changes in oil viscosity, which is mostly affected by the oil temperature. The TCTR and the possible operating parameters can be seen in Figure 5. The modified parameters are marked in red. The mechanical construction described in Section 2.2.1 was extended with a hydraulic unit consisting of an insulated oil reservoir, two hydraulic pumps, two heating elements, a cooling unit, and an electrical pressure control valve, and is located under the experimental area. The oil passes a metal filter before feeding the journal bearing to prevent damage from metal particles. For further information regarding the temperature control system refer to [20].

The possibility to increase the oil inlet temperature $T_{oil,in}$ of up to $100\,°C$ made it possible to use an oil of higher viscosity. A lubricating oil from Addinol (CKT 68) of ISO VG class 68 was used. Due to this change, the pressure application also had to be changed, since higher forces were required to generate mixed or dry friction. The modified hydraulic cylinder can offer up to $F_{N,hydr.} = 20\,kN$.

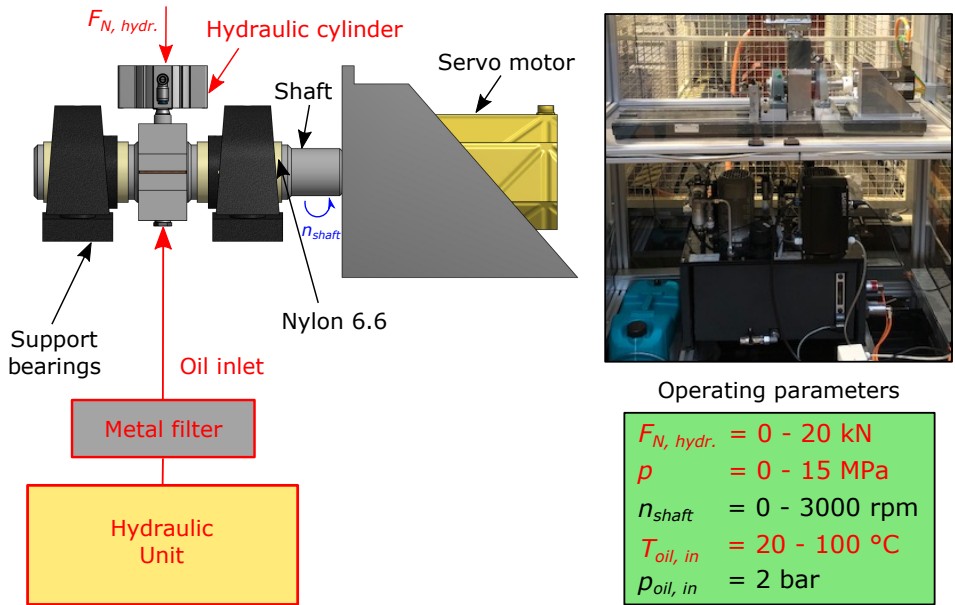

**Figure 5.** Temperature-controlled journal bearing test rig (TCTR) and operating parameters.

At this modified test rig it was further possible to measure the contact voltage (CV) between shaft and bearing. The CV measurement was used in Section 3.1.5 to validate the assumed friction states. A voltage source provides $U_0 = 5\,V$ input voltage. $R_c$ represents the total resistance of the test

components such as the journal bearing and the shaft and is about $0.2\,\Omega$. The resistance of the oil $R_{oil}$ is variable and changes with the state of friction. The CV is measured over the resistance $R$, which has a value of $50\,\Omega$. In fluid friction, the journal bearing and the shaft are separated by a lubricating film. Due to the low electrical conductivity of the oil, a very high resistance is produced at $R_{oil}$. The voltage drops over this resistance so that the CV measured over $R$ is zero. In dry friction, the journal bearing and the shaft come into contact, so that the resistance $R_{oil}$ is almost zero. Nearly the entire voltage can then drop over $R$. The electrical diagram of the CV measurement can be seen in figure 6.

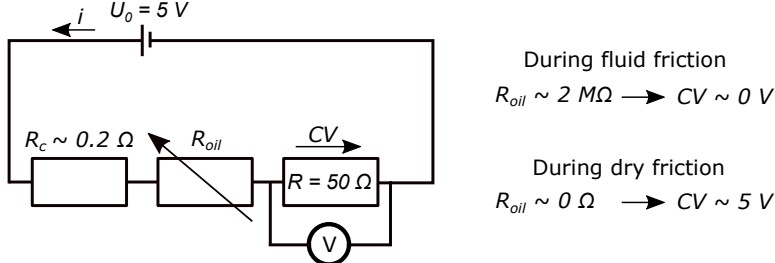

**Figure 6.** Electrical diagram for contact voltage (CV) measurements.

### 2.2.3. Acoustic Emission (AE) Measurement Equipment

To measure occurring acoustic emissions, the broadband piezoelectric Physical Acoustics Corporation (PAC) Wideband (WD) sensor with a frequency range of 100–900 kHz was used (see Figure 7). Epoxy was used as coupling material to the journal bearing back. Since the signal amplitude generated during friction is mostly in the range of microvolts, an amplifier with an integrated bandpass filter (2/4/6 preamplifier) was used. It amplifies the signals into a range of millivolts or volts, thus improving the signal-to-noise ratio. The three amplifier stages 20, 40 and 60 dB can be set, whereby 60 dB was selected for this work because friction signals have a particularly low signal amplitude. A band-pass filter was used to attenuate very low- or very high-frequency noise signals. The bandwidth of this filter is 20–1200 kHz. The output of the pre-amplifier is a voltage in the range of 10 Vpp. This analogue output signal was connected to a 16 bit high-speed analog-to digital (A/D) measuring card from Spectrum (MX4963), which can provide a sampling rate of up to 50 MS/s. The sampling rate was set to 2 MS/s for the AE measurements.

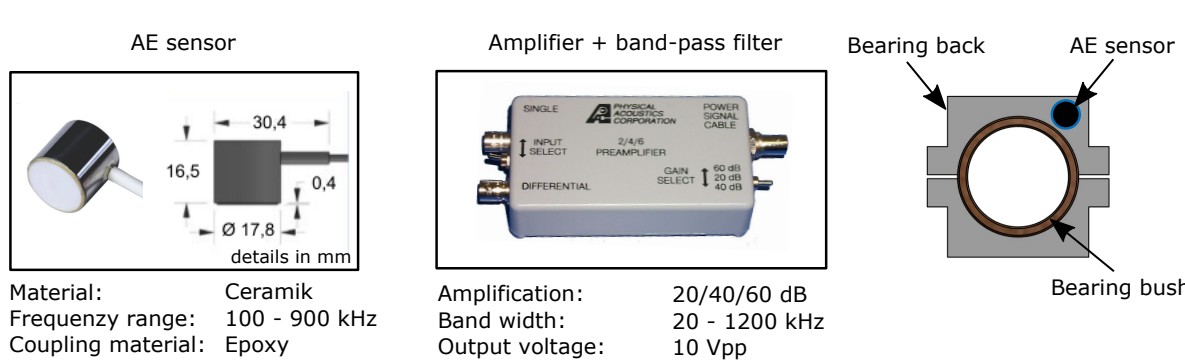

**Figure 7.** Acoustic Emission (AE) sensor, pre-amplifier and sensor position.

### 2.3. Experimental Procedures

Different sets of experiments were carried out: friction experiments (Section 2.3.1), to develop the friction state classifier and to determine the friction localization and wear experiments (Sections 2.3.2 and 2.3.3), to monitor the run-in and long-term wear.

### 2.3.1. Generation of Different Friction States

The primarily requirement for these experiments was to generate all three friction states in order to create a sufficient training data set for the friction state classifier. Mixed and dry friction is in most cases generated during low rotational speeds, high loads or low oil viscosities. In this work first of all the rotational speed and load were varied, later also oil viscosity changes were investigated.

To define suitable operating points DIN 31652 [21] was used for the calculation of the minimum lubricating film thickness $h_{min}$ at varying rotational speeds, radial load and oil viscosity combinations. The critical minimum lubricating film thickness $h_{min,crit}$ indicates the transition to mixed friction and is defined as follows [21]:

$$h_{min,crit} = (0.5...\underline{0.75}...1.0) \cdot (R_{z,shaft} + R_{z,bearing}). \tag{1}$$

If $h_{min}$ is below $h_{min,crit}$ the journal bearing will experience mixed or dry friction otherwise fluid friction occurs. With this information suitable rotational speed ranges can be defined for each load and oil viscosity level so that the journal bearing can experience all three friction states. It should be noted that the calculation according to DIN 31652 cannot indicate the exact transition to mixed friction because of simplifications such as the neglect of surface smoothing by friction. This calculation should only support the selection of suitable operating points.

**Speed and Load Combinations**

Several speed ramps under different radial loads were conducted at the STR described in Section 2.2.1. Figure 8 shows the test procedure and the determined rotational speed-load combinations with regard to the minimum lubricating film thickness $h_{min}$.

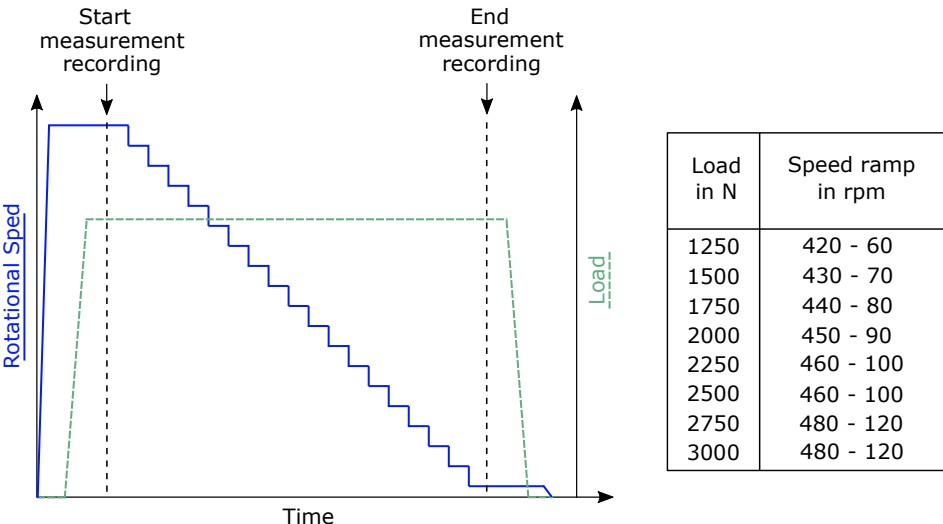

**Figure 8.** Test procedure and determined operation points for speed ramps at constant loads.

**Speed, Load and Temperature Combinations**

As some of the TCTR conditions such as oil type, mean surface pressure or the possibility to control the oil temperature have changed in comparison to the previous setup, the experiments conducted at the STR have to be updated. Figure 9 shows the test procedure and the determined rotational speed-load-oil inlet temperature combinations with regard to the minimum lubricating film thickness $h_{min}$. The data acquired from these experiments were not used to train the classifier, but to investigate the possibility of differentiating the three friction states with AE features under varying speeds, loads and oil temperatures.

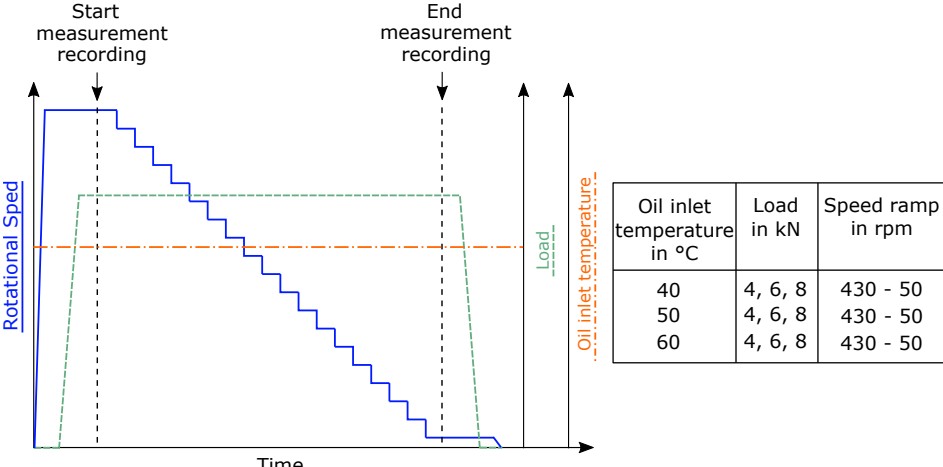

**Figure 9.** Test procedure and determined operation points for speed ramps at constant loads and oil inlet temperatures.

### 2.3.2. Generation of Run-in Wear

The primary requirement for these experiments was to operate in mixed friction conditions for a defined period of time in order to generate run-in wear. Furthermore, the risk of seizure in dry friction should also be minimized by selecting suitable operating points. For these experiments the STR described in Section 2.2.1 was used.

Investigations by Meier [19] have shown that a journal bearing repeatedly "rescues" itself to fluid friction by smoothing the surface roughness when operating without a change to system configuration during the run-in period. Thus, a continuous mixed friction condition within the run-in period is not possible without a change to system configuration. The conditions must be adapted in order to generate further run-in wear. For this reason, Meier [19] defined a short-term test procedure consisting of stationary speed stages and subsequent speed ramps. The test scheme used in this work can be seen in Figure 10.

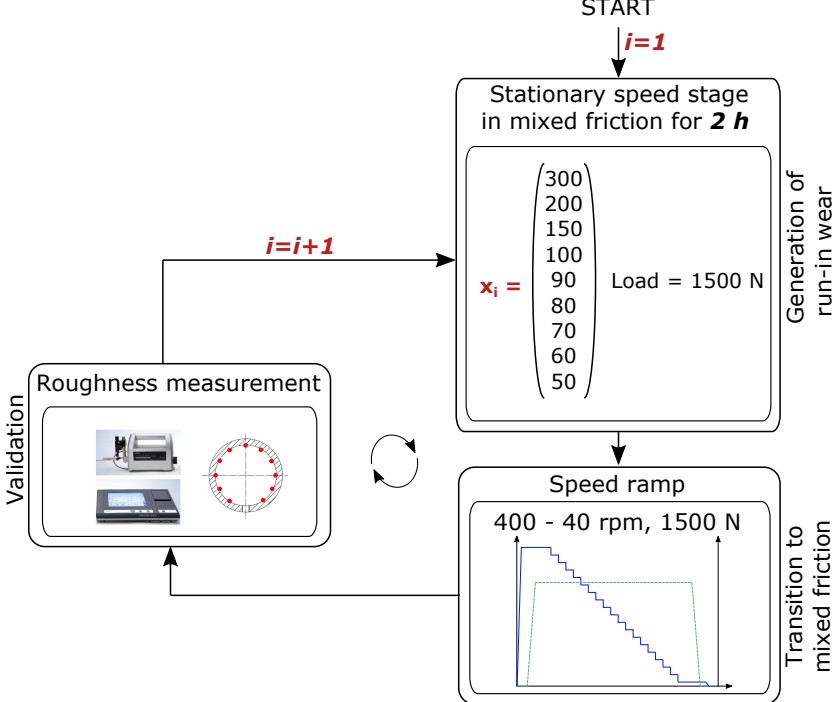

**Figure 10.** Test procedure for short-term tests.

By gradually reducing the speed the larger gap between journal and shaft, caused by smoothing, is reduced, so that mixed friction can occur again [22]. In this work the stationary speed stages were held for 2 h. After each speed stage, a speed ramp was run to indicate the transition to mixed friction. After each speed ramp the journal bearing was dismounted and the surface roughness was measured tactilely. The measurements were carried out for eleven positions distributed around the circumference with a distance of 30 °. A total of nine speed stages were carried out so that the journal bearing was run for a total time of 18 hours.

### 2.3.3. Generation of Long-Term Wear

The primary requirement for these experiments was to operate in mixed friction condition for a long period of time in order to generate long-term wear. Furthermore, the risk of seizure in dry friction should also be minimized by selecting suitable operating points. For these experiments the TCTR described in Section 2.2.2 was used in order to hold the rotational speed, load and oil inlet temperature almost constant over a test period of 18 hours.

After every test period the bearing was removed and the roundness and roughness was measured tactilely. The bearing was then remounted and operated for further 18 h at constant operating conditions. The long-term wear experiments were stopped once a total testing time of 162 h was reached. AE signals were measured every five minutes for a period of 20 s. The experimental conditions are shown in Table 2.

**Table 2.** Experimental conditions for long-term wear tests.

| Number of Measurement | Rotational Speed in rpm | Load in kN | Temperature in °C | Testing Time in h |
|:---:|:---:|:---:|:---:|:---:|
| 1 | 400 | 8 | 60 | 18 |
| 2 | 300 | 8 | 60 | 18 |
| 3 | 200 | 8 | 60 | 18 |
| 4 | 150 | 8 | 60 | 18 |
| 5 | 100 | 8 | 60 | 18 |
| 6 | 80 | 8 | 60 | 18 |
| 7 | 70 | 8 | 60 | 18 |
| 8 | 65 | 8 | 60 | 18 |
| 9 | 55 | 8 | 60 | 18 |

## 3. Results and Discussion

In this section, the results of friction and wear monitoring based on AE and machine learning algorithms are presented. Journal bearing friction monitoring is divided into two main sections:

- Classification of the three main friction states by using machine learning algorithms applied on AE signals (Section 3.1).
- Mixed friction localization over the circumference of the journal bearing by using the AE modulation effect generated during friction (Section 3.2).

Journal bearing wear monitoring refers to:

- Investigations of run-in wear (Section 3.3) by using AE features and tactile measurements as validation.
- Investigations of long term wear (Section 3.4) by using AE features and tactile measurements as validation.

### 3.1. Classification of Journal Bearing Friction States

The procedure for developing a friction state classifier refers to the pattern recognition chain shown in Figure 2. Before a classifier can be trained, signal processing steps are necessary: signal pre-processing to maximize the signal-to-noise ratio or the extraction and selection of separation

effective features. The results of these steps are presented in the following. Afterwards the outcome of the Support Vector Machine (SVM) classifier is shown.

### 3.1.1. AE Signal Pre-Processing

Pre-processing of the acquired AE signals is an important step for successful feature extraction. Without pre-processing, the useful signal, which is in our case the friction signal, is overlaid with noise from other components. The windowing and filtering of the AE signals as central pre-processing steps will be presented in the following.

**Windowing**: For successful feature extraction the AE data should be segmented in such a way that only one friction state class exists within a signal pattern. If windowing is not done there may be an uncertainty in the assignment of the actual friction state class, since multiple friction states could be present in a single set of data. In this work windowing of one shaft revolution was done to avoid different friction state classes in one signal pattern. Figure 11 shows windowed AE signals of the three different friction states generated by reducing the rotational speed n.

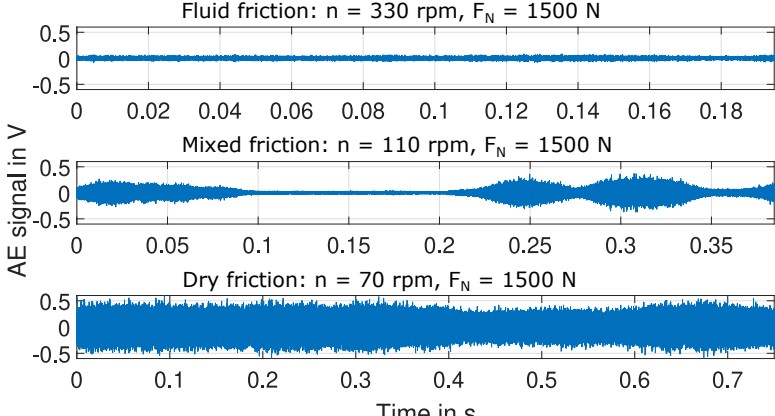

**Figure 11.** Plot of the windowed AE signal over time for fluid friction (top), mixed friction (middle) and dry friction (bottom).

**Filtering**: During operation, the AE signals emitted by friction are overlaid by other machine noises, which makes the subsequent classification of friction states extremely difficult or even impossible. The advantage of using the ultrasonic range for friction detection was already shown by other authors [2]. Machine noises tend to be in the audible range, whereby friction is also detectable in the high-frequency range. The use of a frequency range of 100 kHz–300 kHz for friction detection has so far been proven effective in the literature [23]. In this work, however, a filter bank of band-, high- and low-pass filters was generated, which is shown in Figure 12, to adjust the frequency range for this application. To evaluate the influence of these filters on the AE signal, the root mean square (RMS) value, which is commonly used as an AE feature in the literature, is used.

Ideally, a separation effective feature should not show any major changes in the area of fluid friction, since no mechanical friction occurs in this area. At the transition to mixed friction, the feature should increase with increasing dry friction part in order to be able to distinguish between fluid friction and the other two friction states. The greater the difference between fluid friction RMS value and mixed or dry friction RMS value, the better the friction conditions can be differentiated. In order to distinguish mixed and dry friction other features are needed to create a feature space.

When applying the low-pass filter LP1 with a cut-off frequency of 50 kHz, an increase in the RMS value with increasing speed scan be seen in the area of fluid friction. One of the reasons for this is the presence of machine noises. The transition area to mixed friction is also considerably damped by machine noises. It is difficult to differentiate between fluid and mixed friction by using this kind of filter. In contrast, the use of bandpass or high-pass filters significantly attenuate the machine noises. In fluid friction area an almost linear curve can be seen and the transition area to mixed friction is

clearly distinguishable from fluid friction. The best result is provided by the digital high-pass filter HP1 with a cut-off frequency of 100 kHz.

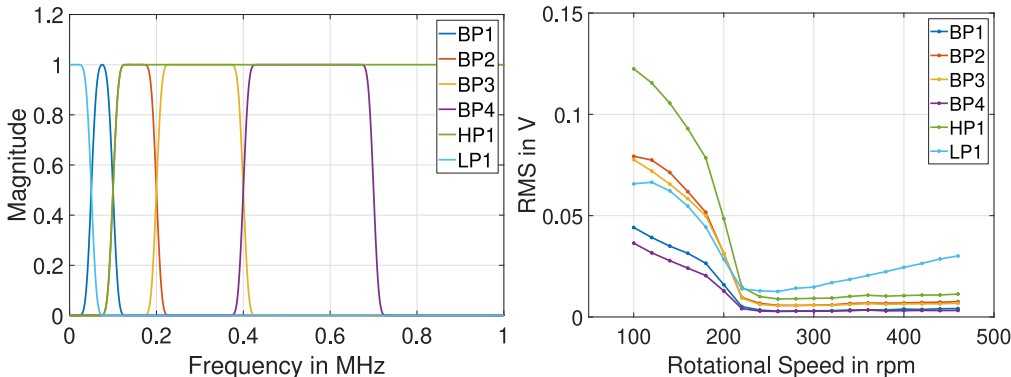

**Figure 12.** Filter bank (**left**) and the AE root mean square (RMS) over the rotational speed at a constant load of 2250 N after signal decomposition (**right**).

### 3.1.2. Feature Extraction

For feature extraction, statistical features such as RMS, skewness, kurtosis, crest factor, clearance factor, Shannon entropy, median frequency etc. were extracted from time, frequency, and time-frequency domain [24,25]. Afterwards, the most separation effective features were used for the classifier. These features were selected manually using a-priori knowledge as the common feature reduction methods could not be applied due to missing labels of the actual friction states. This manual procedure allows a small number of features, which can also be checked for plausibility. Afterwards, the selected separation effective features were used to label the data, which was done by k-means clustering (Section 3.1.3).

**Time domain features**: Equation (2) shows the calculation of the RMS, where $y(n)$ is a signal series and $N$ represents the number of data points.

$$\text{RMS} = \sqrt{\frac{\sum_{n=1}^{N}(y(n))^2}{N}} \tag{2}$$

Figure 13 shows the RMS curves extracted from the windowed and 100 kHz high-pass filtered AE signals for speed ramps at constant loads of 1250 N and 1750 N. The analogy to the Stribeck curve is clearly visible. As the speed decreases, the RMS value also decreases, whereby a minimum is reached at 120 rpm for a load of 1250 N. From this speed on the RMS value increases again. It is quite obvious that this minimum could indicate the transition to mixed friction (transition MF). Furthermore, it is known from the theory of the Stribeck curve that this minimum shifts towards higher speeds with increasing load. This can also be seen in the RMS of the AE signal when compared with the load 1750 N. The transition to dry friction becomes visible by combining several features. This combination is illustrated in Section 3.1.3.

It should be noted that the marked border to mixed friction is an assumption based on the change of the feature at this speed. The assignment of the feature patterns to the respective friction states is done by a clustering procedure (described in Section 3.1.3), since no validation value was available at this point.

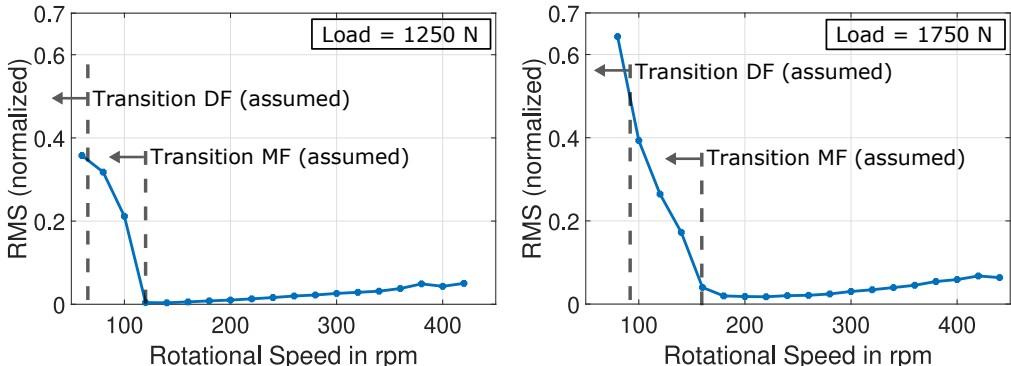

**Figure 13.** RMS of the windowed and high-pass filtered AE signal over the rotational speed at a constant load of 1250 N (**left**) and 1750 N (**right**).

In Equation (3) the calculation of the kurtosis is shown:

$$\text{Kurtosis} = \frac{\sum_{n=1}^{N}(y(n) - \text{Mean})^4}{(N-1) \cdot \text{StandardDeviation}^4} \tag{3}$$

$$\text{Mean} = \frac{\sum_{n=1}^{N} y(n)}{N} \tag{4}$$

$$\text{StandardDeviation} = \sqrt{\frac{\sum_{n=1}^{N}(y(n) - \text{Mean})^2}{N-1}} \tag{5}$$

Figure 14 illustrates that the kurtosis does not show any significant change with decreasing speed until the transition to mixed friction. When reaching the transition to mixed friction the roughness peaks come into contact causing the AE patterns start to change in shape (peakedness is generated). The kurtosis starts to rise for that reason. A maximum is reached and the kurtosis starts to decrease as the roughness peaks are removed and sliding contact becomes stronger. The shift of the transition area to higher speeds at higher loads can also be seen for this feature.

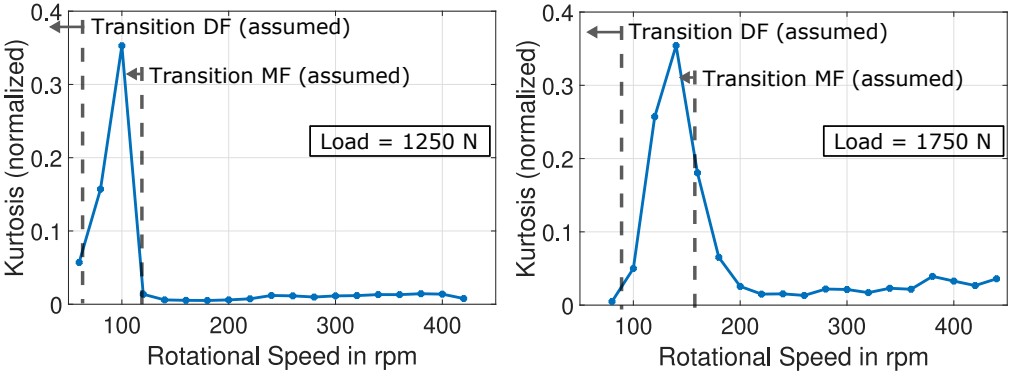

**Figure 14.** Kurtosis of the windowed and high-pass filtered AE signal over the rotational speed at a constant load of 1250 N (**left**) and 1750 N (**right**).

The Shannon Entropy is calculated as follows:

$$\text{Shannon Entropy} = -\sum_{n=1}^{N} y(n)^2 \cdot \log(y(n)^2) \tag{6}$$

It describes the information content of a signal. If no new information is contained in the signal, the entropy does not change. If, on the other hand, further information is created, the entropy increases. Figure 15 shows the similarity of the entropy curve to the RMS curve. They differ, however, especially

in the area of fluid friction. The entropy curve is much flatter there, which makes it much easier to distinguish between fluid and mixed friction.

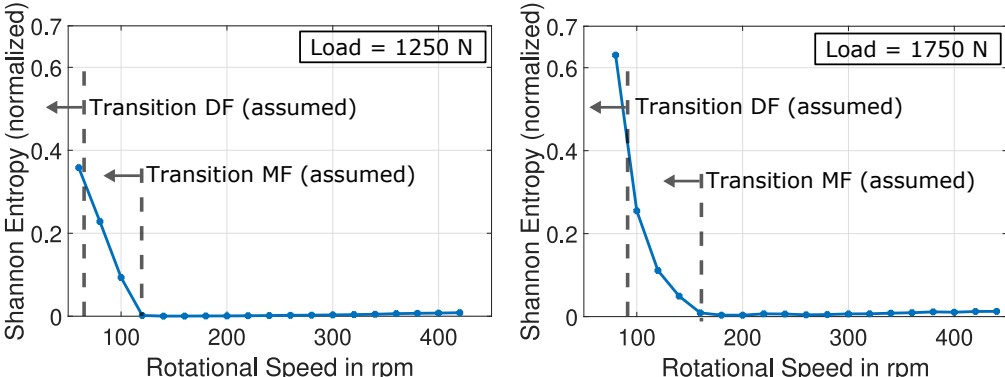

**Figure 15.** Shannon Entropy of the windowed and high-pass filtered AE signal over the rotational speed at a constant load of 1250 N (**left**) and 1750 N (**right**).

The Shannon Entropy seems to be, compared to the RMS, more separation effective for the classification of the friction states.

**Frequency domain features**: The median frequency $f_{MedF}$ was extracted from the frequency range:

$$\int_{f_1}^{f_{\text{MedF}}} f \cdot U(f)^2 df = \int_{f_{\text{MedF}}}^{f_2} f \cdot U(f)^2 df \tag{7}$$

It describes the frequency at which the power spectrum is divided into two regions of equal amplitude. It is also described as half the total power. Figure 16 shows the Median Frequenzy plotted over the rotational speed for different constant loads. At the transition to mixed friction, the median frequency seems to increase suddenly and remains almost constant with decreasing speed. This is due to the increased influence of high-frequency signal components on the power spectrum during mixed and dry friction. This feature seems be effective for differentiating fluid friction from the other two frictional states.

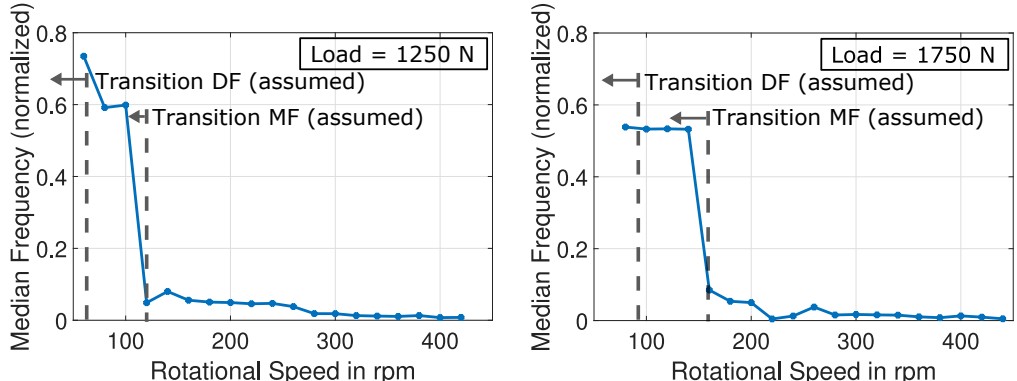

**Figure 16.** Median Frequency of the windowed and high-pass filtered AE signal over the rotational speed at a constant load of 1250 N (**left**) and 1750 N (**right**).

**Time-frequency domain features**: In addition to the features extracted from time and frequency domain, features from time - frequency domain were also extracted and evaluated. Continuous wavelet transform (CWT) was used for this purpose. The Morlet Wavelet was used as the mother wavelet in this work.

To obtain separation effective features, the wavelet coefficients should be determined for small scales. The scales between 6–16 were determined, since these correspond to pseudo frequencies of approx. 100–270 kHz for the Morlet Wavelet. To evaluate the CWT, compared to high-pass filtered

signals, features were extracted from identical operating points. Figure 17 shows that the use of CWT slightly improved the separation efficiency of each feature.

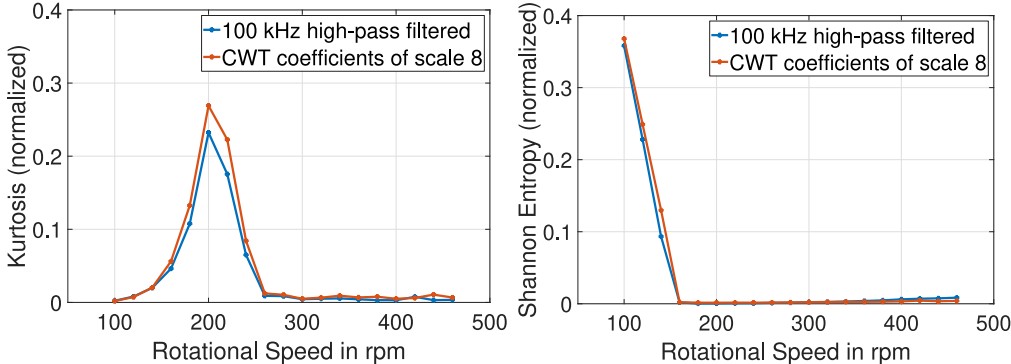

**Figure 17.** Comparison of kurtosis (**left**) and Shannon entropy (**right**) each for 100 kHz high-pass filtered and continuous wavelet transform (CWT) coefficients of scale 8 for a speed ramp at a constant load of 2250 N.

### 3.1.3. Data Labelling

A clustering procedure became necessary in order to label the data because of missing validation values. Such methods are used for example in unsupervised learning and are therefore suitable for applications where the assignment of a data series to the correct class is not known. The method used in this work is the k-means algorithm. The pattern $x_n$ is assigned to a predefined number of clusters $k$ in such a way that the sum of the squared distances of each pattern to its cluster center of gravity $\mu_k$ is minimal. The variable $r_{nk} \in$ describes a set of binary indicator variables. It is $r_{nk} = 1$ if $x_n$ is assigned to cluster $k$, otherwise $r_{nk} = 0$. Therefore, the values for $r_{nk}$ and $\mu_k$ must be selected in such a way that the following function is minimized [26]:

$$J = \sum_{m=1}^{N} \sum_{k=1}^{K} r_{nk} ||x_n - \mu_k||^2. \tag{8}$$

The friction state patterns to be labelled were derived from the combination of kurtosis, Shannon entropy and median frequency. In Figure 18, the clustered feature patterns grouped with k-means clustering are shown in color.

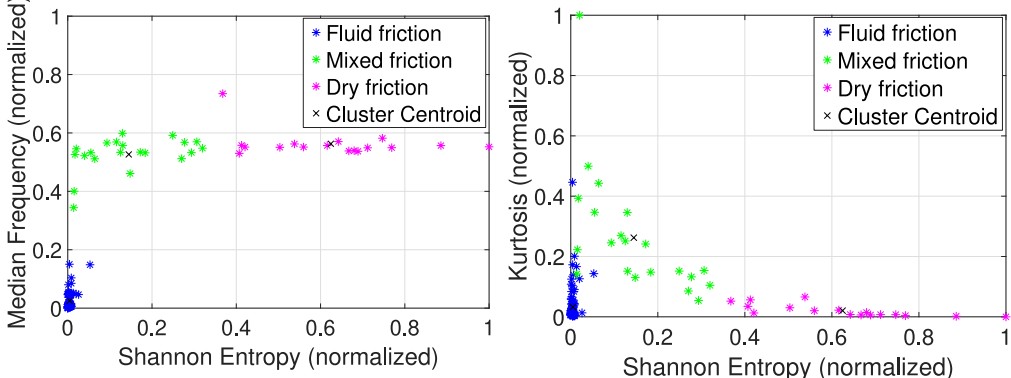

**Figure 18.** Clustered feature patterns with k-means clustering.

### 3.1.4. SVM Classifier

The SVM classifier trained with this data set achieved an overall detection rate of 96.7% for the three-class problem. The efficiency of the classifier for the classification of the different friction states is shown by the confusion matrix in Figure 19. A detection rate of 90% was achieved for mixed friction, 10% of the feature patterns were wrongly assigned to dry friction. With this training data set and

with this procedure of labelling, pre-processing and feature extraction a detection rate of 100% could be achieved for fluid and dry friction. For further investigations the data should be labelled with a validation value such as friction torque or contact voltage (CV). Since this set of data was acquired at the STR no validation measurement was possible. The trained classifier can now be applied to new and unseen friction data.

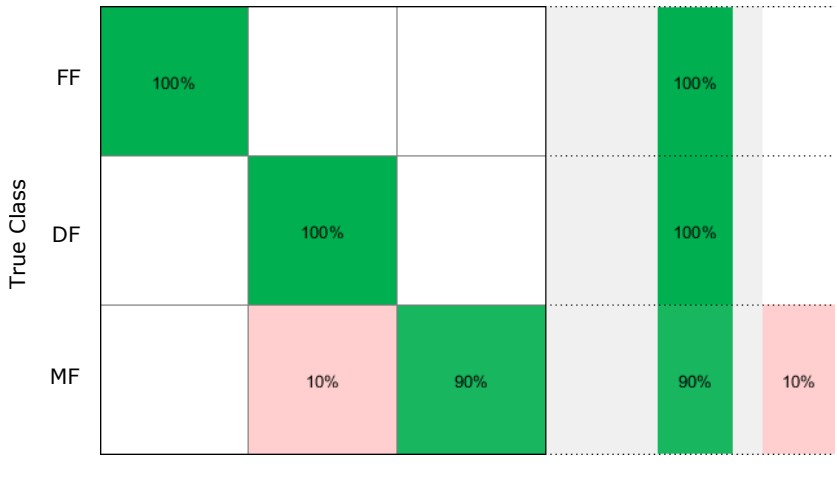

**Figure 19.** Result of the trained Support Vector Machine (SVM)-classifier with cross validation.

### 3.1.5. Influence of Temperature Variations

Figure 20 shows the CV over the rotational speed for different oil inlet temperatures at constant loads of 6 kN and 8 kN. It is known from the Stribeck curve that with increasing oil temperature the oil viscosity decreases and therefore the supporting characteristic of the oil decreases as well. The consequence is that the transition area from mixed to fluid friction is shifted to higher rotational speeds.

If there is no mechanical contact between shaft and journal bearing, the CV is zero. With decreasing speed, the roughness peaks of the two sliding partners come into contact at a certain transition speed, so that the CV begins to rise. The maximum CV is reached when the shaft and the journal bearing are fully in contact. Furthermore, it can be seen that the transition to mixed friction (transition MF) moves towards higher speeds with increasing oil temperature. This agrees with the theory that a decrease in oil viscosity leads to lower supporting properties of the oil.

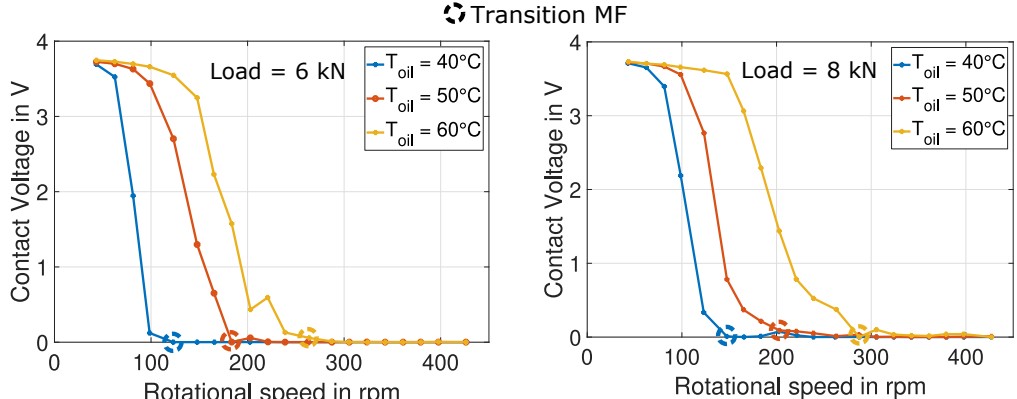

**Figure 20.** CV over the rotational speed for different oil inlet temperatures at a constant load of 6 kN (**left**) and 8 kN (**right**).

Figure 21 shows the AE feature Shannon entropy over the rotational speed for different oil inlet temperatures at constant loads of 6 kN and 8 kN. The expected flat curve in the area of fluid friction

can be seen for all curves. With decreasing speed the transition speed to mixed friction is reached and the curves start to rise. This observation also agrees with the results obtained for speed and load combinations. It can further be seen that the amplitude in mixed friction increases with increasing oil temperature. This indicates the detectability of the influence of the lower oil viscosity on the friction force with this AE feature. Furthermore, the amplitude of this feature is lower for a load of 6 kN compared to a load of 8 kN. This could indicate the lower friction force generated for lower loads in mixed friction, under otherwise identical operating conditions.

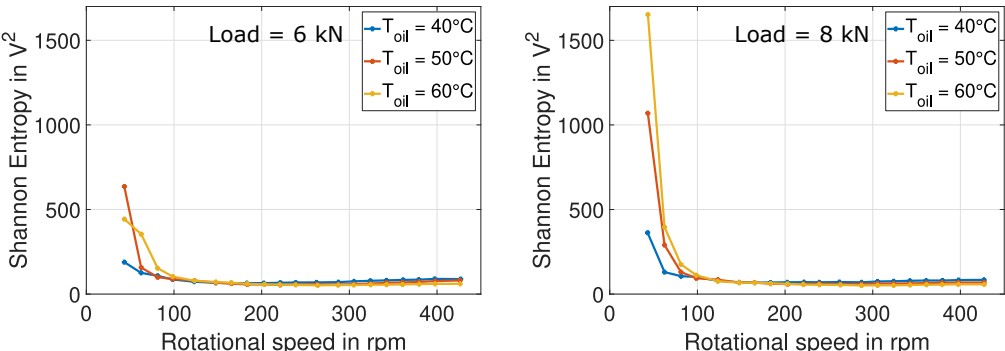

**Figure 21.** Shannon Entropy over the rotational speed for different oil inlet temperatures at a constant load of 6 kN (**left**) and 8 kN (**right**).

To illustrate the transition speeds to mixed friction for this feature, Figure 22 is used. It shows a detailed section. The CV was used as validation value to mark the transitions to mixed friction. It can be seen that for the same load, the minimum of the curves moves towards higher speeds with increasing temperature. It can also be observed that the feature in the area of fluid friction, although flat, has a higher amplitude for lower oil temperatures. A higher oil viscosity causes higher resistance in fluid friction and thus a higher hydrodynamic coefficient of friction. However, the amplitude in the range of fluid friction is low enough that it should not have a significant influence on the differentiation of mixed and fluid friction. This feature thus provides plausible results even under variable oil viscosity, caused by different oil inlet temperatures.

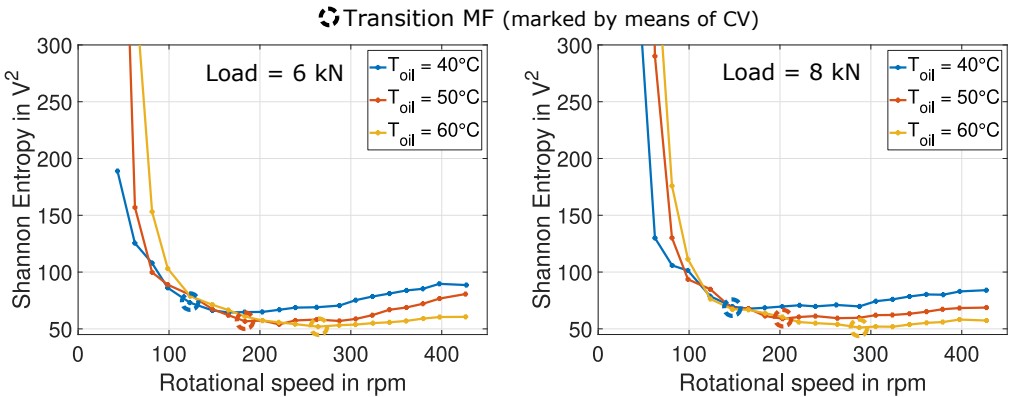

**Figure 22.** Detail section of the Shannon entropy over the rotational speed for different oil inlet temperatures at a constant load of 6 kN (**left**) and 8 kN (**right**).

Figure 23 shows the kurtosis of the AE signal. Again, the transitions to mixed friction have been marked using the CV. As expected, the kurtosis has a relatively flat trend in fluid friction. At the transition to mixed friction the kurtosis starts to increase. Furthermore, shortly after the transition, a further decrease in kurtosis can be observed, followed by a significant increase. The reason for this could not be conclusively clarified.

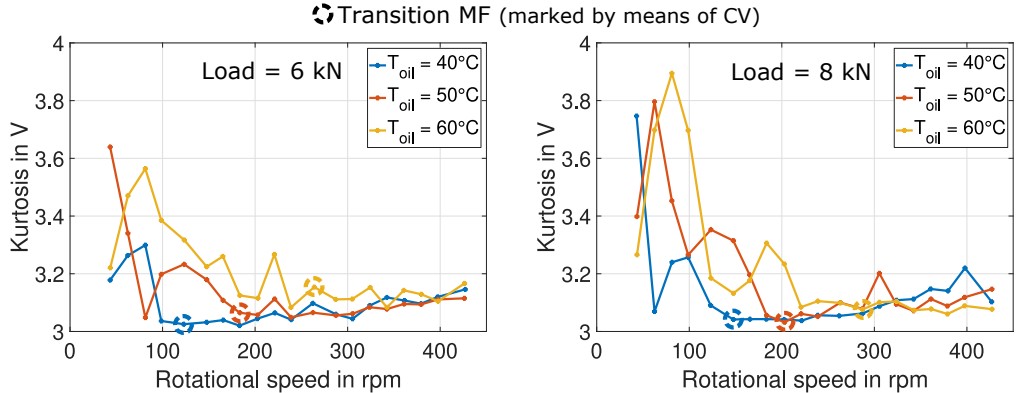

**Figure 23.** Kurtosis over the rotational speed for different oil inlet temperatures at a constant load of 6 kN (**left**) and 8 kN (**right**).

## 3.2. Localization of Journal Bearing Mixed Friction Events

Although the developed classifier can differentiate between the different friction states, a statement about the friction position $\phi$ or friction distance $s_R$ is not possible. This knowledge is significant for applications where friction does not occur at the same location, such as for non-stationary load zones or shaft imbalances. These parameters provide important information for determining the remaining useful lifetime (RUL) of a journal bearing. Repeated friction at the same position reduces the lifetime more than friction events of the same number distributed over the circumference. The accumulation of mixed friction events at the same circumferential position can thus be interpreted as a measure of the journal bearing coating wear. Furthermore, $s_R$ is directly related to the wear volume $V_w$ by the following equation:

$$V_w = k_w \cdot F_R \cdot s_R, \tag{9}$$

where $k_w$ represents the specific wear rate or wear intensity and must be determined experimentally and $F_R$ the friction force.

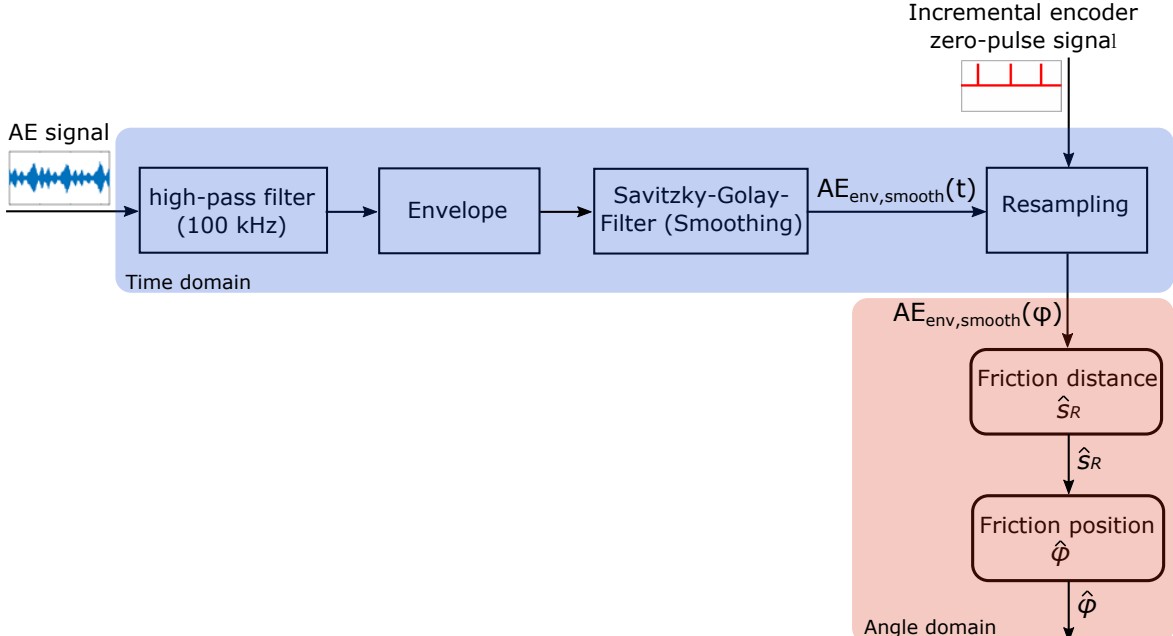

**Figure 24.** Procedure for localization of mixed friction events.

In Figure 11 a clear amplitude modulation can be seen within mixed friction state. Other authors [27,28] already detected this modulation effect in seals and interpreted it as occurring friction

events. In his work Hall [27] determines the friction position of different seals along a shaft by using this AE modulation effect. Inspired by this work, the idea is to use this modulation effect to determine $\phi$ and $s_R$ over the bearing circumference. Figure 24 shows the procedure of this localization method.

### 3.2.1. Envelope Curve and Smoothing in Time Domain

In the first step, all values of the high-pass filtered AE signal smaller than zero were set to a value of zero. The envelope curve was then generated to determine the local minima and maxima. Methods such as the Hilbert transformation can be used to create the envelope. For the application presented here it is sufficient to determine the energy of the envelope because it is only necessary to know where local maxima and minima occur. The method used here calculates the RMS value over a certain number of signal points and saves this value in a new vector. This creates a kind of moving average of the signal, from which the local maxima and minima can be determined. However, this envelope is affected by some noise, so that this curve was smoothed for further determination of the local minima and maxima. Low-order approximation polynomials are suitable for this purpose in order to achieve the best possible smoothing. One possibility is to use the Savitzky-Golay-Filter. This method smooths a signal by fitting a polynomial function piecewise to the signal. The local minima and maxima can now be clearly determined from this new curve.

### 3.2.2. Resampling to Angle Domain

In the next step, the smoothed envelope is transferred from time to angular domain. The zero-pulse signal is used for this purpose. From a time $t_1$ to a time $t_2$ exactly one rotation and thus $2\pi$ has passed. Between the peaks, the missing angles are interpolated in such a way that the same number of samples is created as the smoothed envelope of the AE signal. Each data point of the AE signal can thus be assigned to an angle. In Figure 25 the smoothed envelope is plotted over the shaft angle.

By defining a certain threshold value on the y-axis, a estimated friction distance $\hat{s}_R$ can be determined. It is the distance between the points of intersection of the threshold value with the envelope. The threshold value must be determined for each application.

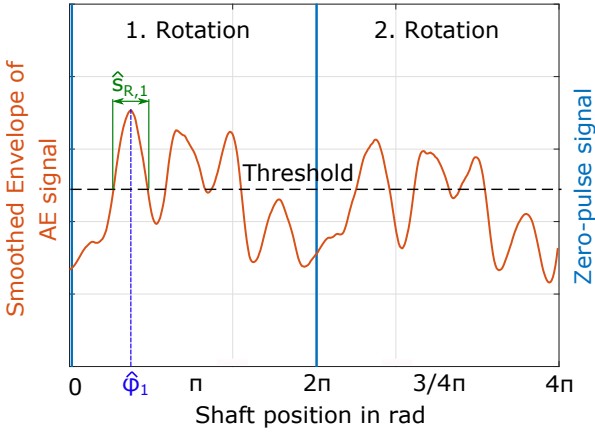

**Figure 25.** Smoothed envelope of the AE signal during mixed friction at 80 rpm and 1500 N over the shaft angle.

To prove this method and to determine a suitable threshold value, the roughness of the bearing surface has to be measured. This allows a validation of the assumed friction distance $\hat{s}_R$ and position $\phi$ as these areas should be smoothed.

### 3.3. Monitoring of Journal Bearing Run-in Wear

For complete monitoring of journal bearings besides the knowledge of the current friction state also the knowledge of wear caused by friction is necessary. The state of health of journal bearings is affected by wear. The shape changes so that the supporting lubricating oil film can no longer be formed above a certain wear level. The total wear of a journal bearing over its lifetime consists of run-in, long-term and progressive wear. The maximum amount of wear is reached at the transition to progressive wear. So, to determine the state of health, the wear level should be monitored up to this stage of wear.

It can be assumed that the AE features already determined for friction monitoring could also be applicable for wear monitoring, since it is known that there is often a proportional relationship between friction and wear. This section shows the possibility of monitoring the run-in wear of a journal bearing using AE technology and the already developed separation effective features. Run-in wear is the amount of wear caused by the smoothing of surface roughnesses. It starts from the beginning of the run-in time with a brand new journal bearing and ends when the contact surfaces have be adapted to each other.

The speed ramps carried out after each speed stage are suitable for evaluating the relationship between feature drift and run-in wear (see Section 2.3.2). These ramps show the transition to mixed friction. In case of identical speed ramps, a shift can only be caused by a change of the bearing surface. Figure 26 shows the Shannon entropy and kurtosis of the AE signals for speed ramps 40–400 rpm and 1500 N after the corresponding stationary speed stages. The feature Shannon entropy, which indicates the transition speed to mixed friction, shifts towards lower speeds as the test time increases. The maximum of the kurtosis, which also indicates the transition to mixed friction, shifts towards lower speeds as well. Only one outlier is visible. The reason for this could be a wear particle in the lubrication gap. The already mentioned smoothing of the journal bearing surface can thus be determined by AE features. Since this set of data was acquired at the STR no validation measurement was possible. The transitions to MF result from the investigations shown in Section 3.1.5, where CV could be used as a validation parameter.

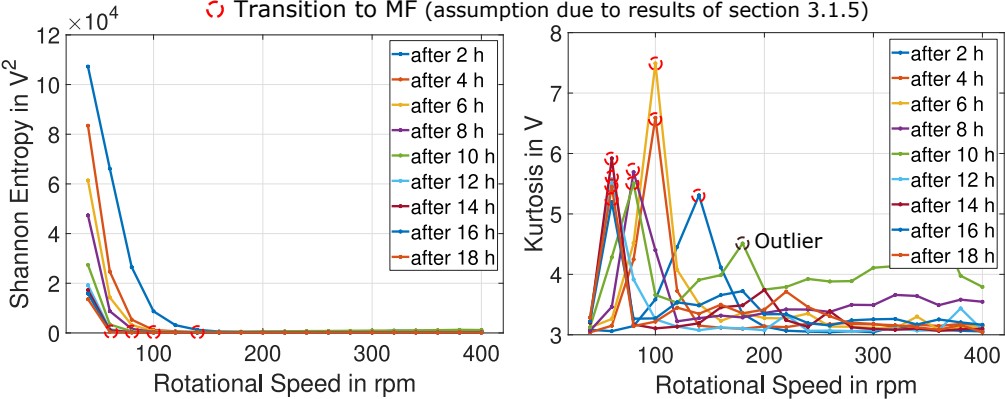

**Figure 26.** Shannon Entropy and Kurtosis of AE signals for speed ramps 40–400 rpm and 1500 N after the corresponding stationary speed stages.

In his work, Meier [19] shows that the coefficient of friction decreases with increasing run-in wear for the same operating point. It is interesting to note that this can also be seen in the feature Shannon entropy. As an example, the speed 40 rpm is mentioned here: The amplitude decreases chronologically with the testing time. For this reason, it can be assumed that this feature cannot only distinguish between friction states, but also indicates a variable correlating with the frictional force $F_R$. This is an important finding for further investigations regarding long-term wear. This would make it possible to determine the actual wear amount due to the proportional relationship between friction and wear.

To validate these results, the averaged mean surface roughness $\overline{R_z}$ of the journal bearing was determined after each speed ramp:

$$\overline{R}_z = \frac{R_{z30} + R_{z60} + ... + R_{z300} + R_{z330}}{11} \tag{10}$$

With the mean surface roughness $R_z$:

$$R_z = \frac{1}{5} \cdot (R_{z,lr1} + R_{z,lr2} + R_{z,lr3} + R_{z,lr4} + R_{z,lr5}). \tag{11}$$

The measuring distance $lr$ is standardized and in this case $lr = 0.8$ mm.

Figure 27 shows the averaged mean surface roughness $\overline{R_z}$ over the testing time. After the first four hours, a significant reduction in surface roughness is visible. After 6 h the roughness seems to increase slightly, but this may be due to metal particles on the surface or measurement uncertainty of the equipment. Afterwards the roughness decreases again.

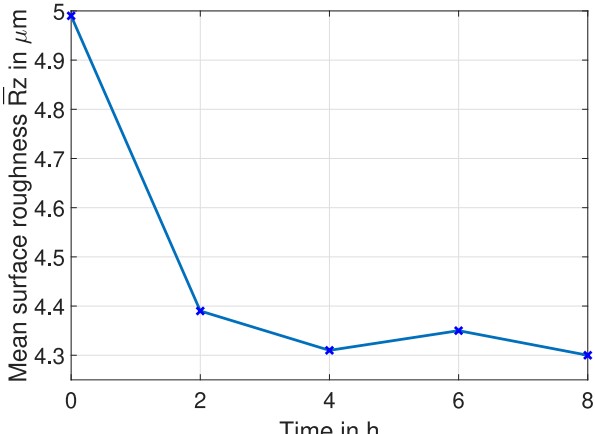

**Figure 27.** Averaged Mean surface roughness $\overline{R_z}$ over the testing time after every test run.

The evaluation of the roughness was stopped after 8 h of testing time, which corresponds to a stationary speed stage of 100 rpm.

*3.4. Monitoring of Journal Bearing Long-Term Wear*

In this section, results of long-term wear investigations are shown. Tactile measurements are used as validation for AE.

After every test run of 18 h the surface roughness was measured in order to determine the smoothing of the surface and to calculate the run-in wear. Figure 28 shows the distribution of the mean surface roughness $R_z$ for the initial state and after a total testing time of 162 h. A strong worn area is clearly visible in the load zone. Outside the load zone a slight smoothing can be seen, which can be attributed to mounting and dismounting as well as to measurement errors. Wear has not occurred continuously over the width of the bearing. The reason for this could be an uneven journal bearing surface on the one hand or the inconsistent pressure distribution over the width of the journal bearing which leads to deformations of the bearing or the shaft on the other hand. To illustrate the progression of roughness over time, the averaged mean surface roughness at an angle of 180° was plotted over the testing time and can also be seen in Figure 28. The surface roughness value starts at $\overline{R_z} = 4.8$ µm and decreases with increasing testing time. At the end of the testing time a value of $\overline{R_z} = 2$ µm is reached.

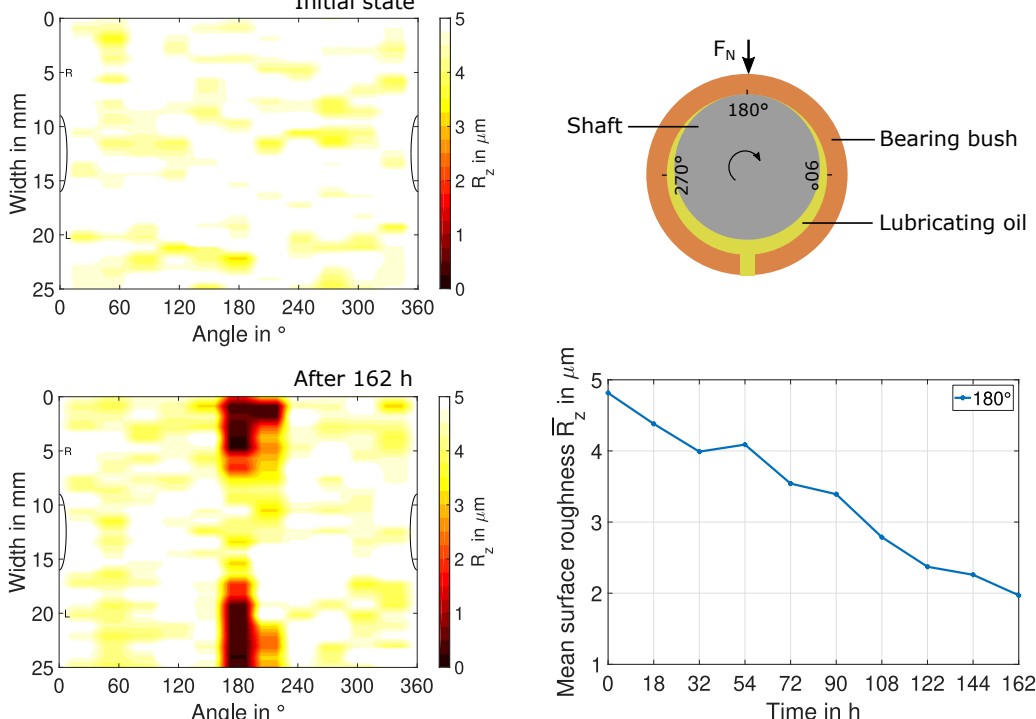

**Figure 28.** Distribution of the mean surface roughness $R_z$ for the initial state and after a total testing time of 162 h (**left**) as well as the averaged mean surface roughness $\overline{R_z}$ at 180° (**right**) over the testing time.

The run-in wear $V_{w,ri}$ caused by the removal of the roughness peaks has only a minor effect on the roundness of the bearing. Since this removal has a significant influence on the AE signal, a method for determining the amount of $V_{w,ri}$ using the roughness measurement data is presented. The roughness profiles $R$ acquired after each wear experiment differ in centerline because the surface conditions are different. To be able to match the profiles and finally calculate $V_{w,ri}$, another reference line must be found. At this point it should be noted that the procedure is based on the assumption that only the peaks are affected by wear and therefore the valleys do not change. First, the median value of all valleys is determined, both for the initial $R_{ini,valleys}$ and wear $R_{wear,valleys}$ measurement. By subtracting these values from each other, an offset is given which is then added to the wear roughness profile data $R_{wear}$:

$$\text{Offset} = \text{Median}(R_{ini,valleys}) - \text{Median}(R_{wear,valleys}) \tag{12}$$

$$R_{wear,new} = R_{wear} + \text{Offset} \tag{13}$$

Thus, the two roughness profiles have nearly the same reference. To determine the wear volume of each $R$-profile, the integral of the positive values is determined and multiplied by the circumference of the bearing. The run-in wear volume $V_{w,ri}$ is then calculated:

$$V_{w,ri} = V_{ini} - V_{wear}, \tag{14}$$

where $V_{ini}$ is the volume calculated from the initial R-profile and $V_{wear}$ from the worn R-profile. Roundness measurements with a tactile measurement device were done in order to calculate the long-term wear volume $V_{w,lt}$. The roundness was measured at six positions distributed over the bearing width. Interpolation was made between the different widths. Figure 29 shows the roundness plot for the initial state and after 162 h. A clear deepening can be seen in the load zone area. This change in shape results from long-term wear.

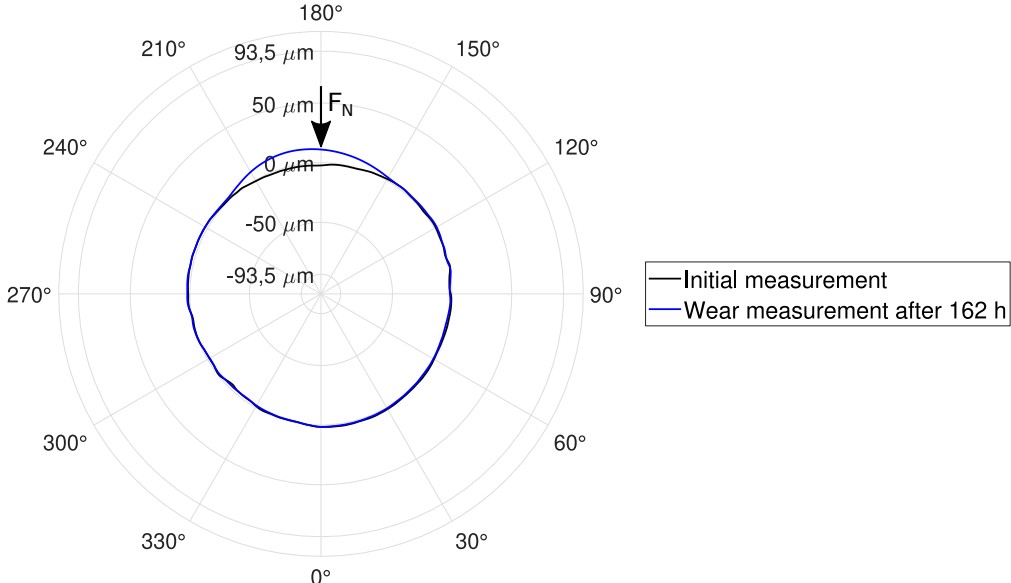

**Figure 29.** Roundness of journal bearing for initial state and after 162 h of testing.

The total wear volume is calculated as follows:

$$V_w = V_{w,ri} + V_{w,lt} \tag{15}$$

Figure 30 shows the total wear volume $V_w$ as well as the integrated AE RMS over the testing time $t$. The wear volume and the AE feature both rise with increasing testing time.

$$\text{Int AE RMS} = \int_0^t \text{RMS}_{\text{AE}} \, dt \tag{16}$$

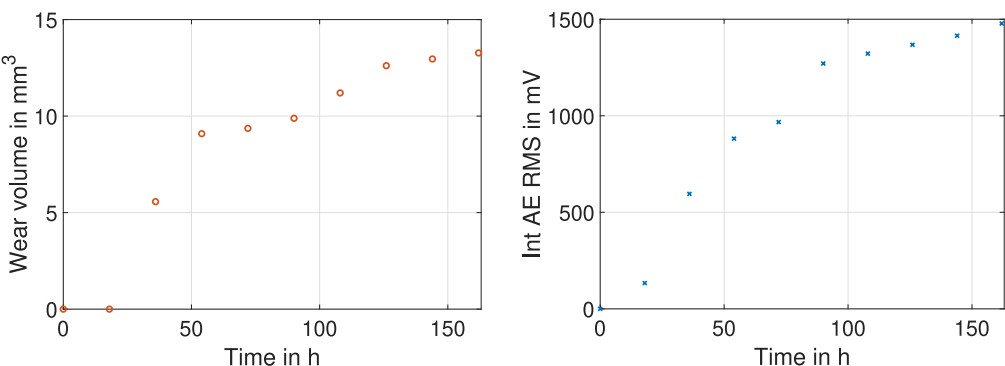

**Figure 30.** Total Wear volume $V_w$ calculated from roundness and roughness measurements (**left**) and integrated AE RMS (**right**) over testing time.

To demonstrate the relationship between these two parameters, the integrated AE RMS was plotted over the wear volume (see Figure 31). The coefficient of determination $R^2$ is 95.78%. This data set shows a good correlation, so that a regression analysis with a larger data set should be done in future. In order to determine the wear volume using only AE features, further wear measurements are necessary. The long-term wear investigations have not been finished at this point. In future, regression algorithms will be applied to estimate the wear condition.

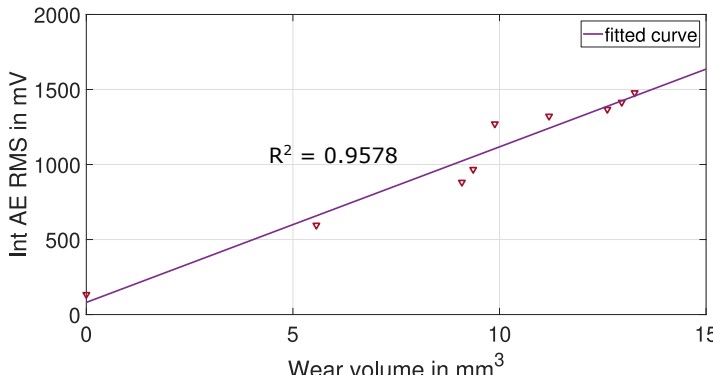

**Figure 31.** Integrated of AE RMS over wear volume.

## 4. Conclusions

In this work, possibilities of monitoring hydrodynamic journal bearings, experiencing friction and wear, and using AE technology and machine learning algorithms were presented.

- Friction state classification: This was done under varying rotational speeds and radial loads by pre-processing the AE signals, extracting and selecting suitable AE features from time, frequency and time-frequency domain using CWT and applying SVM as classifier. A feature vector consisting of the features Shannon entropy, kurtosis and median frequency was the input for the classifier. An overall detection rate of 96.7% was achieved for this three class problem. Furthermore, it was shown that it is possible to distinguish the three friction classes with AE even under different oil viscosities.
- Mixed friction localization: This was done over the circumference of the bearing by making use of the AE modulation effect. The envelope of the AE signal was smoothed and fused with the zero-phase signal of an incremental encoder to resample it from time to angle domain. The local maxima show the friction position $\phi$ and by adding a threshold the friction distance $s_R$ can also be determined.
- Monitoring of run-in wear: Short-term wear test were done to monitor the run-in wear with the use of separation effective AE features. With increasing run-in wear there was a clear shift visible in the AE features. These results were validated with tactile measurements of the journal bearing surface.
- Monitoring of long-term wear: Long-term wear investigations were done. There is a correlation visible between the wear volume and the integrated AE RMS but further research is needed in this area.

## 5. Further Work

In parallel to this work a wireless power and data system has been under development to ensure AE sensors can be emplaced within the planet carriers of the planetary gearbox and that good signal could be obtained in the necessary frequency bands despite the very high amplitude noise from gear mesh and other sources that would be expected in this environment. The system was originally developed under an EASA project [14] and has since undergone further refinement to improve performance in specific implementations.

Figure 32 shows the wireless system arrangement. The transceiver A uses 12 V DC power from a switch mode power supply and a coax connection for the sensor telemetry. It is connected to a coil matching network (B) which is case mounted and provides good impedance match between the driving system and the coil while not being such a good match as to reduce signal bandwidth potential between the matched coils. The coil on the rotating side is connected to a power harvesting system (C) which also includes the sensor amplification and filtering system to support the AE sensor

(D). Available power on the rotating side is limited to about 10 mW but is sufficient for acquisition, amplification, filtering, and transmission needs of the current sensor.

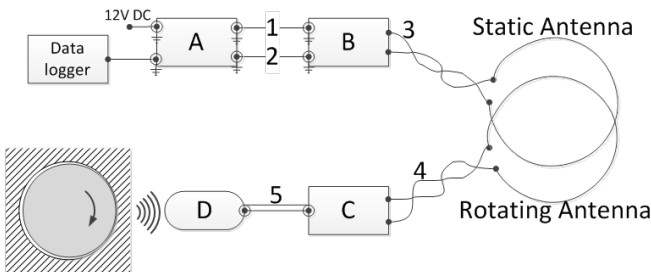

**Figure 32.** WDTU system arrangement.

A sample of the test results shown in Figure 33 shows the stability of the signal at 277.85 kHz as the test rig decelerates and the strength of the signal of interest (green trace in bottom plot) at about 40 db above the local noise floor (red trace). The areas of the spectrogram from which these traces are taken are indicated by the dotted green and red lines in the middle plot which is the relevant portion of the spectrogram. The top plot shows the rpm of the test rig during this time.

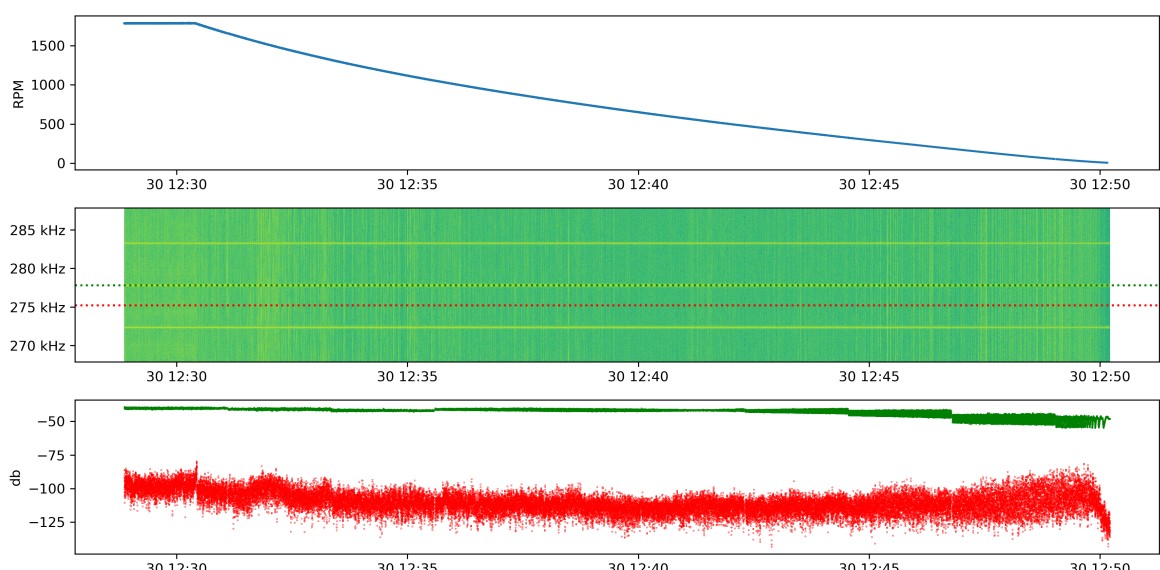

**Figure 33.** Test rig RPM profile, signal spectrogram section, and signal and noise comparison from testing.

The joint application of the developed journal bearing monitoring methods together with the WDTU on a planetary gearbox is the topic of investigations at Technische Universität Munich chair of FZG in 2020.

## 6. Patents

Nowoisky, S.; Mokhtari, N. Method and Device for Monitoring a Slide Bearing. in *European Patent Office*, EP3447469, **23.08.2019**.

Nowoisky, S.; Mokhtari, N.; Pelham, J.G. Method and Device for Monitoring a Journal Bearing. DE 10 2018 123 025.7, **19.09.2018**, pending Patent.

Nowoisky, S.; Mokhtari, N.; Grzeszkowski, M. Verfahren und Vorrichtung zur Schätzung des Verschleißzustandes eines Gleitlagers. DE 10 2018 123 571.2, **25.09.2018**, pending Patent.

Nowoisky, S.; Ciciriello, L.; Grzeszkowski, M.; Mokhtari, N. Method and system for detecting a functional failure in a power gearbox and a gas turbo engine. EP19194753.0, **30.08.2019**, pending Patent.

**Author Contributions:** Conceptualization, S.N. and N.M.; Data Curation, N.M., J.G.P., S.N. and J.-L.B.-G.; Methodology, Software, Validation, Formal Analysis, Investigation, Resources, N.M., J.G.P. and S.N.; Writing—Original Draft Preparation and Visualization, N.M., J.G.P. and S.N.; Writing—Review & Editing, N.M., J.G.P., S.N., J.-L.B.-G. and C.G.; Supervision and Project Administration, S.N. and C.G. All authors have read and agreed to the published version of the manuscript.

**Funding:** Part of this research project was funded by the Federal Ministry for Economic Affairs and Energy LAROS (20T1510).

**Acknowledgments:** Part of the content of this paper was derived from the research project LAROS (20T1510). This project is embedded in the national civil aviation research program. The aim of this project is to develop new bearing technologies for aircraft engines. Furthermore, we acknowledge support by the Open Access Publication Funds of TU Berlin.

**Conflicts of Interest:** The authors declare no conflict of interest.

## Abbreviations

The following abbreviations are used in this manuscript:

| | |
|---|---|
| A/D | Analog-to-digital |
| AE | Acoustic emission |
| BPR | Bypass ratio |
| CV | Contact voltage |
| CWT | Continuous wavelet transform |
| EASA | European Union Aviation Safety Agency |
| DF | Dry friction |
| FF | Fluid friction |
| FFMEA | Function Failure Mode & Effect Analysis |
| FZG | Forschungsstelle für Zahnräder und Getriebebau |
| Int | Integrated |
| MF | Mixed friction |
| PAC | Physical Acoustic Cooperation |
| PGB | Power Gearbox |
| RFID | Radio Frequency Identification Device |
| RMS | Root Mean Square |
| RPM | Revolutions per minute |
| RUL | Remaining useful lifetime |
| STR | Small journal bearing test rig |
| SVM | Support Vector Machine |
| TCTR | Temperature-controlled journal bearing test rig |
| WD | Wideband |
| WDTU | Wireless Data Transfer Unit |

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
