# Peer review of "Friction and Wear Monitoring Methods for Journal Bearings of Geared Turbofans Based on Acoustic Emission Signals and Machine Learning"

_lubricants, doi:10.3390/lubricants8030029_

Round 1

Reviewer 1 Report

This is a massive work and the authors should be congratulated for a great work. It is a very long text and it includes several sub-studies. The authors should (but mustn't) consider to split it into 2-3 manuscripts. 

In general this is a very timely and very good work. I hope it can be a kind of reference work for this area in the future. However, I do want to ask the authors to discuss or improve some few things:

Line 183, eq. 1: This is a new expression for non-German readers (I think). It needs some clarification. The limits, 0.5, 0.75, and 1.0 what do they mean? Is is the transitions between BL, ML and FFL? If so, only two values are needed. What is the third one? 

Figure 11 and text: It is already here assumed that the three sets of data corresponds to the three regimes. How do you know? 

Figure 13: It is very interesting to see that RMS of the signal increases rapidly when speed is decreased. It is natural to assume that this corresponds to the transition to Mixed lubrication. On the other hand, in the right figure at load 1750 N the authors didn't choose the point where the signal start to increase but a somewhat higher speed. Is that deliberate? Why?

Figure 14: is even more interesting. The kurtosis seems to have a maximum at lower speed than the assumed transition to mixed lubrication. What does it illustrate? Does it have a meaning, please explain the mechanisms for that. Is that a sign of wear and the peakedness goes down when peaks are removed due to the smoothening of the surface? One should also remember that there is a fourth regime in between mixed lubrication and full film lubrication. That is the micro-EHL regime which are given in some textbooks (see for example Hamrock's book). The micro-EHL regime is a full film regime with higher friction than the minimum value in the Stribeck curve. Still full film but the pressure is probably peaky since load is carried by asperities that form small EHL contacts. It might be so that the transition in Fig 13 is the transition to micro-EHL while the peak in Fig 14 corresponds to the transition from "peaky" micro-EHL to mixed. Please discuss this.

Section 3.1.3 is unclear. How was it possible to classify a point as "dry friction"? The authors haven't discussed how they detect distinguish dry friction from the mixed lubrication regime. Nothing in the RMS or kurtosis seems to tell about that. Please explain. 

Figure 20 and text. No information is given about how the contact voltage was measured. Please add that to the description of the experimental setup. 

Figure 20: How do you know that 1 volt corresponds to transition to mixed lubrication? The signal starts to rise rapidly already at 0.1 volt. Sp why setting 1 volt as the threshold?

Figs 21 and 22: It is not clear to me what the differences are between these two figures. Maybe one can be removed?

A general question: What is causing the AE-signal in full film lubrication? The viscous friction is pretty gentle. And why does it increase with speed? 

Eq. 9: Is a little bit unusual wear equation where friction is part of it. The authors claim several times that wear and friction follows each other, but is that really true? In ploughing and adhesison this can be true but when wear debris come into play it may lower friction even if there is wear. And there might be chemical wear that leaches the material (selective transfer) with high wear as result. But low friction. The authors need to jsutify this equation and their claim. For this application. 

Line 453: "By defining a certain threshold ...": How is the threshold chosen?

Line 472: What is a "clear running mirror"? I can understand what the authors mean but the English needs to be improved. 

General: No info is given about the position on the bearing of the AE sensor. Is it on the housing or the bearing shell? And at what angle?

Fig. 26: The Shannon entropy is used as evidence of reduced friction. But are there any supporting friction measurements as evidence? Or is it just an assumption?

Fig. 28: No info about roughness measurement methodology/technique is given. How i Rz defined? It must be defined over a limited sampling length/area? Please explain.

Line 523: It is sound to assume that valleys remain unchanged but they might be filled with debries etc. Meaning that also valleys can be shallower in the same way as peaks become lower. Have you checked this?

Reviewer 2 Report

In the paper the results of the experimental activity performed by the authors aimed at the estimation of wear condition and wear volume is well described.

Anyway the paper is a bit confused about the test rigs and need to be modified

all the results and the paper are based on the first test rig and its improvement with temperature control. The reviewer suggest to avoid to stress about the presence or the temperature control and other issues. The absence of such device is used by the author for explaining the lack of some measurement. But it is not a real problem. the “second” is the WDTU test rig that is only introduced for describing wireless setup and issue, but it is not used for the results except the last section 3.5 in which a simulated signal have been used. Furthermore the authors at pag. 3 also said that the real application will be based on rolling element bearings in which the methods used in the paper probably cannot be applied and different approach need to be used.

Therefore it is suggested to remove the part of WDTU (section 2.2.3, 3.5 and conclusions), and modify the title by removing the words “of geared turbofans”. The paper is well focused on wear of journal bearing. It is also suggest to move all the WTDU and wireless part into a new paper, maybe to be submitted to a more specific journal such as “Sensors” from the same editor.

Other comments:

It is better to add the nominal and the real clearance of the bearing at the beginning o the activity In figure 3 the roughness Rz is 4um whereas in the experiments starts from approximately 5um. Please fix them In figure 3 the bearing width is 40mm but in the results is 25mm (figure 28) In figure 4 the specific pressure range is 0-1.5Mpa but in the results the pressure range up to 4MPa. Also the load increased from 3KN to 20kN (table 4). What modification in the test rig have been introduced? In table 1 the viscosity is 68 but in the text (test rig without improvements) is 10. What is the right one? In the test rig nylon rings are used to damping the system. In a real harsh environment, does the method described will be valid (RMS , kurtosis, etc?). Please comment this point Eq 6. “Shannon Entrop”IE” At pag 14, CWT is introduced but it is not necessary and can be removed. What do the authors want to demonstrate? Scale 8 appears in the legend of figure 17 and scale 3 in the caption. What is the right one? Some time the load in the paper appears in kN and MPa. Please standardize it. The contact voltage CV is used. Do the authors measured also the capacity and/or the resistance? For the results in figure 25 maybe it is better to average the signal over more revolutions. At pag. 21 the author explain the occurrence of a not uniform wear along the axial direction to the uneven bearing surface. It is also due to two aspects: i) the pressure distribution of journal bearing, that is maximum in the middle, that leads to deformation(compression) of the bearing in the middle and than the contact occurs at the two bearing side. Ii) the deformation of the shaft and than the contact mainly occurs again close to the sides. The discussion about the centerline at pag 21 is not clear. Try to explain better How the roundness has been measured? Fig 29. Please rotate the figure of 90° CCW in the direction of the load and add an arrow indicating the direction of the load. What is the integrated AE RMS. How is it obtained?

Round 2

Reviewer 1 Report

I suggest publishing

Author Response

The reviewer suggests publishing.

Reviewer 2 Report

The authors modify the paper. Anyway for my point of view the last section 3.5 must be removed. It is far from the topic of the paper and its content is useless.

Author Response

Point 1: Anyway for my point of view the last section 3.5 must be removed. It is far from the topic of the paper and its content is useless.

Response 1: We removed section 2.3, 2.4.4 and 3.5 and added chapter 5 as a short discussion chapter for further work, which links the AE work to the need for a wireless system to be used within planetary gearboxes.